# Biliary Atresia Animal Models: Is the Needle in a Haystack?

**DOI:** 10.3390/ijms23147838

**Published:** 2022-07-16

**Authors:** Nutan Pal, Parijat S. Joy, Consolato M. Sergi

**Affiliations:** 1Jefferson Graduate School of Biomedical Sciences, Thomas Jefferson University, Philadelphia, PA 19107, USA; nutanpal17@gmail.com; 2Department of Internal Medicine, University of Iowa Hospitals and Clinics, Iowa City, IA 52242, USA; parijat-joy@uiowa.edu; 3Anatomic Pathology Division, Department of Laboratory Medicine and Pathology, Children’s Hospital of Eastern Ontario, University of Ottawa, Ottawa, ON K1N 6N5, Canada; 4Department of Lab. Medicine and Pathology, Stollery Children’s Hospital, University of Alberta, Edmonton, AB T6G 2B7, Canada

**Keywords:** biliary atresia, liver, congenital, perinatal, animal model

## Abstract

Biliary atresia (BA) is a progressive fibro-obliterative process with a variable degree of inflammation involving the hepatobiliary system. Its consequences are incalculable for the patients, the affected families, relatives, and the healthcare system. Scientific communities have identified a rate of about 1 case per 10,000–20,000 live births, but the percentage may be higher, considering the late diagnoses. The etiology is heterogeneous. BA, which is considered in half of the causes leading to orthotopic liver transplantation, occurs in primates and non-primates. To consolidate any model, (1) more transport and cell membrane studies are needed to identify the exact mechanism of noxa-related hepatotoxicity; (2) an online platform may be key to share data from pilot projects and new techniques; and (3) the introduction of differentially expressed genes may be useful in investigating the liver metabolism to target the most intricate bilio-toxic effects of pharmaceutical drugs and toxins. As a challenge, such methodologies are still limited to very few centers, making the identification of highly functional animal models like finding a “needle in a haystack”. This review compiles models from the haystack and hopes that a combinatorial search will eventually be the root for a successful pathway.

## 1. Introduction

The biliary system is one of the most complicated systems of the human organism [1]. It entails an extrahepatic and an intrahepatic component, which seem to differentiate separately [1]. Both components eventually join in a unique excretory system of bile drainage. The extrahepatic biliary system results from a process intimately associated with pancreatic embryogenesis [2]. Conversely, the intrahepatic component derives from the periportal areas of the ductal plate, and the co-localization of polyductin with liver progenitor cell markers is critical [3,4]. The ductal plate is the prototype of embryogenesis [5,6]. If the joining of the intra- and extrahepatic components fails, such disconnection results in atresia of the biliary drainage system (BA) [1,3,4,5,7]. Biliary atresia is a very complex disorder and, in the most recent years, the adjective fibro-obliterative has slowly replaced the previously accepted term, necro-inflammatory [8,9,10]. Bove et al. reported detailed histology in 172 centrally reviewed biliary remnants performed at Kasai hepatic portoenterostomy (KHPE). Active lesions were classified as either necro-inflammatory or active concentric fibroplasia with or without inflammation, whereas inactive lesions disclosed bland replacement by collagen and fibrous cords with little or no inflammation. We need to acknowledge that heterogeneity is a common and substantial finding in BA. According to Bove et al., it did not correlate with age at KHPE, congenital anomalies at laparotomy, and outcome. Prevalence of partially preserved epithelium in active fibroplastic lesions may indicate that epithelial damage (regression or injury) may not be a primary event. Alternatively, the re-epithelialization is already in progress at the time of KHPE. The unfathomable balance between active fibroplasia and re-epithelialization of retained, collapsed but not obliterated lumens is remarkably intriguing. At places, it may explain the heterogeneity of BA histopathology reports [11].

It has also been suggested that BA is the consequence of repeated bouts of the inflammation destined to clear the toxins derived by the malfunction of the bile drainage system [3,7,10]. The liver hilum’s surgical and pathological inspection shows, indeed, an obstructive cholangiopathy. There is an annihilation of the lumen and progressive iterative cycles of inflammation with an overwhelming NLRP-3 cytokine response [10,12,13,14]. BA may include *situs viscerum inversus* and is labeled BA splenic malformation syndrome (BASM). Its clinical presentation typically occurs in the 2nd week after birth [15,16]. A variant of BA is the cystic BA (CBA), which has shown a better outcome than the classic embryonic and fetal types and does not seem to require liver transplantation.

The search for the perfect animal model goes hand in hand with the quest to identify the etiology of BA. Substantially, there have been two lines of thought: one school has deposited trust in a viral etiology, whereas another school sustains that genetic mechanisms are probably the underlying disorders with or without the influence of toxins. The existence of viral-induced BA in mice has been often extrapolated to suggest that viral-mediated injury must play a role in human BA, but extensive human data have failed to support such a hypothesis. In the viral etiology, there is the concept that an immune or autoimmune response to the perinatal infection of a virus may lead to necrosis of the biliary epithelium. Cytomegalovirus (CMV) [17,18,19,20], reovirus [21,22,23], rotavirus [21,24], Epstein-Barr virus [21,25,26], or human papillomavirus (HPV) [20,21] can induce a subsequent inflammation and/or accompany an evidence of inflammation. Such infection is followed by a progressive fibro-obliteration of the lumen of the interlobular bile ducts. Some culpable viruses have been published in case reports only, but extensive human data have failed to validate such a hypothesis, which was never substantially confirmed as causative in case-control studies or cohort studies. The ductal plate malformation (DPM), which is the archetypical defect of the primitive biliary system, has been identified in some cases with BA associated with splenic anomalies [4,7,27,28,29,30,31,32,33,34,35]. The complexity of the degree of inflammation and state of the development of the biliary system in BA, combined with the urgent need for new therapeutical approaches, requires well-defined animal models that enable the translation of research findings to clinical use.

## 2. Results


**Experimental Animal Models Conundrum**


Some rodent models are well suited for studying mechanisms of various aspects of inflammation. Still, they have shown limited utility for translating genetic research data to human terms, thereby impeding therapeutic developments. There is a need to emphasize the need for standardization of preclinical animal models by showing the impact of duration of jaundice, sex, and methodologies. Recognizing and realizing the limitations and advantages of preclinical models will assist a more effective translation of experimental results to enhanced approaches to therapy for human BA. This neonatal cholangiopathy proceeds through several overlapping stages that unequivocally necessitate precise spatiotemporal control. It also needs contributions from multiple cell types that orchestrate the myriad of cellular processes that eventually result in BA. This extraordinary process is evolutionarily conserved and can be studied in numerous in vivo models, ranging from zebrafish to primates. Over the past several decades, the use of BA models has contributed to breakthroughs, advancing the knowledge and interpretation of the molecular and cellular events that contribute to this process. Despite advances in the biology of understanding BA, understanding the pathophysiology of acute and chronic inflammation in the developing liver is limited, resulting in a lack of successful therapies for the affected infants.

Animal models that have been designed to investigate biology and preclinical mechanisms resulted in variations that have not been properly standardized and validated, creating a challenging situation. Several variables influence this already complex biologic process. They include age, sex, biliary development, metabolic pathways, and microbiome diversity. Consequently, we encounter a conundrum. On the one hand, we face the situation where the field is required to use animal models to perform preclinical investigations as a necessity for therapeutic development. On the other hand, we are often confronted with the limited predictive value of data obtained in preclinical examination for clinical implementation. It is critical to understand both the limitations and advantages of various models. Accurate experimental animal models may allow the understanding of pathophysiological mechanisms involved. Different animal species like the rat, lamprey, and rhesus monkey have been used, but murine models have been used more often. The reproduction of BA requires various strategies, including surgical methods (e.g., common bile duct (CBD) ligation), viral infection, grafting, and drug administration. We will describe the available animal models, their development, applications, and practical considerations for scientific usage.


**Non-Human Primates and Non-Primates Occurrence of BA**


Spontaneously occurring BA-like findings have been reported in lambs, dogs, calves, and foals but have not been adopted for studies. The more investigated spontaneous BA models in animals are zebrafish, lamprey, and Rhesus monkey and animal models for BA can be subdivided in viral, surgical, toxin-based, and genetic (Figure 1).


**Zebrafish**


Although distant from the human condition, the tropical freshwater zebrafish (*Danio rerio*) may be used as a model for studying biliary defects due to the remarkable conservation of development genes. Since the 1960s, the zebrafish model has become increasingly important to scientific research. This tropical fish native to Southeast Asia has many characteristics that make it a valuable model for studying human genetics and disease. The complete genomic sequence of the zebrafish was published in 2013, and its genome reaches 1,505,581,940 base pairs in length (26,247 protein-coding genes) [36,37,38]. The *notch* pathway, which is essential in mammalian biliary development, also regulates the specification of liver progenitor cells towards a biliary cell fate in zebrafish [39]. In the zebrafish model of BA, the inhibition of DNA methylation leads to biliary defects and activation of interferon gamma (IFN-γ) genes. Activating the IFN-γ pathway is sufficient to cause developmental defects. Subsequently, there is a decrease in cholangiocyte proliferation and expression of *vhnf1* (*hnf1b*, *tcf2*) [40,41,42]. In addition, BA patients show, in the cholangiocytes, a reduction of the levels of apical glypican-1 (GPC1) [43,44,45,46,47,48,49], a heparan sulfate proteoglycan involved in regulating hedgehog signaling. Knockdown of *gpc1* in zebrafish leads to biliary defects, which can be partially reversed with cyclopamine, a hedgehog antagonist. Most recently, *HNF6*, *ARF6*, *Sox17*, *ILF2*, *DNMT1* genes have been confirmed to have effects on the development of bile ducts in zebrafish [46,50,51,52,53,54,55,56,57,58]. In particular, the deletion of *ilf2* determined a reduction of bile flow and defect in the biliary anatomy, which has been interpreted as a phenocopy of BA in humans. It is imperative to investigate the embryonic biliary development in Zebrafish in more detail and compare BA development as induced by virus (RRV) or toxins (biliatresone) in heterozygous/homozygous *ilf2* knockout rodents to further illuminate the roles of *ILF2* in the human embryonic biliary development and BA [50].

In Table 1 are detailed the most paramount animal models of biliary atresia.


**Lamprey**


The Sea Lamprey (*Petromyzon marinus*) is a primitive, eel-like fish native to the northern Atlantic Ocean and freshwater of the Great Lakes [59,60,61,62,63]. The sea lamprey metamorphosis is a genetically pre-programmed animal model for BA [64,65,66,67,68,69,70,71,72,73]. The larva has a fully formed biliary system with the gallbladder, lost during the transition to the adult stage. The epithelium of the extrahepatic ducts in lamprey larvae transforms and expands into a caudal portion of the pancreas of the adult. Regression of the biliary structures is accompanied by periductular fibrosis. Although Youson initially noted these morphological similarities in 1978 [66,74], the lamprey was not adapted as an experimental BA model until recently in this century [59,64,65,67,68,69,70,71,72,73,75]. In human BA, the most severe injury seems to occur early in the extrahepatic tract, and the intrahepatic ducts show ductular proliferation. In the lamprey, the degenerative process is asynchronous. The more rapid decline occurs in small peripheral components, and post-metamorphosis lampreys usually grow to adult size without developing any sort of progressive disease, despite having no bile ducts. Yeh et al. reported that the sea lamprey adapts by de novo synthesis and secretion of bile salts in the intestine and reduction of bile salt synthesis in the liver [71]. According to Cai et al. [70], adult sea lamprey tolerates BA by transforming its bile salt composition from toxic C24 bile acids (petromyzonol sulfate) to less harmful C27 bile acids (3-keto-petromyzonol sulfate). It accelerates renal excretion through urine, supported by marked upregulation for organic anion and bile acid transporters. Further studies may be necessary to explore, validate, and leverage this BA model.


**Rhesus Monkey**


Rhesus monkey represents a vital compromise to have an animal model that is close to humans and more feasible for surgical manipulation [76]. The only pure extrahepatic biliary atresia known in a nonhuman primate is identified in the rhesus monkey (*Macaca mulatta*) neonates. In 1983, Rosenberg et al. showed clinical and pathologic similarities to BA in a six-week-old female monkey [77]. Jaundice and conjugated hyperbilirubinemia at the age of six days persisted throughout life with the portal proliferation of biliary structures. The autopsy revealed biliary cirrhosis with ductular proliferation at ten months. High reovirus 3 titers in rhesus neonates’ serum were also found in human infants with BA [23,78]. In 1974, Landing first hypothesized that hepatotropic viruses play an etiologic role in BA, but numerous studies have substantially annihilated the viral hypothesis. It is important to keep in mind that no detailed studies have been performed on the rhesus neonate animal model due to the costs and difficulty in obtaining Ethics Committee approval.


**Experimental Models of BA**



**Surgical Models**



**Bile Duct ligation (BDL)**


Since 1932, with the first publication by Cameron and Oakley [79], BDL in rats was used as a classic BA model [80,81,82]. In 1988, Medeiros et al. have reported different histology secondary to CBD obstruction in adult and young rats [83]. In 1997, Omori et al. [84] used a BDL model to compare tissue expression of alpha-fetoprotein during the developmental process and found it to be more intense in young animals and, indeed, absent in adults. Gibelli et al. [85] evaluated the BDL effects in Wistar rats aged six days and adults. The initial response of ductule proliferation and inflammatory infiltrate were less intense in the newborn animal, but the portal and periportal fibrosis were significantly higher when compared with adult animals. These results were quite similar to those obtained by Medeiros et al. and Omori et al. but are different from those obtained by the Swiss group. The Swiss study has shown that there is an increase of fibrogenesis in adult animals with a decrease of liver parenchyma compared to the young organ. Thus, BDL may be a valid model using neonatal rodents. Ductular proliferation and fibrosis occur temporally independently in young animals and fibrogenesis may be the result of losing regulatory mechanisms at the intercellular level. In fact, the young animal showed a deficiency of collagenases and metalloproteases responsible for fibrotic tissue degradation. Further studies should target the point at which the evolution of fibrosis to cirrhosis becomes irreversible. Microsurgical cholestasis is claimed to be better than the simple BDL in terms of lesser incidence of clinical complications. In this microsurgical technique, resection of the extrahepatic biliary tract, including the CBD, occurs in continuity with the bile ducts that drain the four lobes of the rat liver. As a drawback of this model, investigations remain technically difficult. They require special anesthetic procedures and microsurgical equipment, and expertise. Maintaining the viability of post-surgical younger animals is also more challenging compared to that of adult mice.


**Rhesus monkeys/CBD/Postnatal/lymphatic drainage**


Devadas et al. [86] partially ligated and then resected the CBD in young rhesus monkeys. After three weeks, the thoracic duct was shunted into the jejunum. Subsequently, there was a spontaneous reconstitution of the bile ducts and relief of jaundice. Although not fully explained, these observations may support the surgical procedures incorporating the lymph drainage into biliary drainage operations. Further studies using Rhesus monkeys may not be feasible due to ethics concerns in several institutions and countries.


**Lamb CBDL model is beneficial to study the cystic type of BA**


Prenatal BDL has been successfully applied to lambs to create a BA model. Spitz [87] carried out intrauterine ligation of the CBD in fetal lambs at approximately 80 days gestational age (survival rate: 60%). The animals were observed at varying intervals up to the 36th day of life. The histopathology of the bile duct lesions resembles at places the extrahepatic form of BA (Figure 1). This model was analogous to the cystic BA in humans but has not been commonly used due to the substantial time required for the experimental setup associated with costs and ethical concerns regarding experiments in higher vertebrates. Pickett et al. [88] reported a similar model of vascular ligation in fetal lambs with similar effects on the bile ducts evident by birth.


**Rabbit may disclose an Intrahepatic Bile Duct Model**


In rabbit fetuses, ligation of the hepatic artery seems to produce relatively selective ischemia of the intrahepatic biliary system. In this model there is an absence or hypoplasia of the intrahepatic biliary system in five-week-old pups [82].


**Transplantation Model**


The morphological changes observed in bile ducts segments that were transplanted into mice have been found to be like those seen in BA. Schreiber et al. [89] grafted segments taken from mice at fetal day 18, postnatal day 7 and day 21, and adulthood (>6-weeks) under the renal capsule of adult mice. After three weeks, prominent fibrosclerosis occurs. Histology such as primary sclerosing cholangitis was evident in the grafts from older donors. Furthermore, the progression of the rejection injury in the adult bile duct grafts was associated with an induction of class I and class II histocompatibility antigen expression in the bile duct epithelium. The severity of this injury could be attenuated by immunosuppression of the recipient.


**Virus-based Models**


A range of hepatotropic viruses has been identified in liver biopsies of BA patients, suggesting a viral infection’s role as the initial trigger for the development of BA. Rhesus rotavirus (RRV)-induced BA models are one of the animal models for experimental BA, which has been favored by some schools.


**Reovirus 3**


Reovirus 3 has been implicated in spontaneous BA in the rhesus monkey, and intraperitoneal injection of it in weaning mice produces chronic obstructive jaundice. The choledochal obliteration is like human BA, but many characteristics of BA are missing in this model; however, there is a high level of subjectivity in interpreting findings. Unlike humans, this BA model can only be induced at weaning time with only transient or segmental obliterative fibrosis [78,90,91,92]. Despite persistent jaundice, early clearance of the virus from the liver and bile ducts occurs. This event may be the reason for the failure of confirmation of this virus in the liver of two- to three-month-old infants with BA. This model has further been used to delineate the importance of some factors, including the sialic acid as co-receptor, the M2 gene, and the S1 gene. All three aspects seem to be relevant for the induction of BA in mice. The poor reproducibility of demonstrating reovirus-3-inducing BA is the reason for this murine model’s low popularity.


**CMV**


A Swedish research group showed CMV antibody levels in mothers of BA infants to be significantly high, and CMV DNA was also detected in some infants with BA [93]. Wang et al., 2011 [94] reported a perinatal CMV-induced hepatobiliary system injury model in guinea pigs with biliary tropism of CMV along with a characteristic inflammatory injury. A Th1 cell-mediated antiviral immune response was found to cause newborn guinea pigs’ hepatobiliary damage. Despite weaknesses, this model may help to explore viral infection as the cause of biliary injury.


**HPV**


Domiati-Saad et al. [20] failed to demonstrate any evidence of infection with HPV types 6, 16, 18, and 33 in BA. Furthermore, there is not a well-followed-up model demonstrating HPV infection in an infant’s liver. Still, the participation of HPV in BA remains probably intriguing and not fully explored.


**Rotavirus**


In 1974, Landing et al. hypothesized that BA could represent the result of a virally induced process of the hepatobiliary tree. Twenty years later, the support of such a hypothesis occurred with the obstruction caused by the oral or intraperitoneal administration of group A rotavirus in the newborn mouse. In 1993, Riepenhoff-Talty reported extrahepatic biliary obstruction in newborn Balb/c-mice infected with RRV [95]. The intrahepatic histology was found to be similar to human BA, at least partially (Figure 1). The prominent morphological findings were long segment atresia (with hydrops of the gallbladder) (37%), segmental atresia of the proximal CBD (16%), segmental atresia of the distal CBD (21%), short atresia with long segment dilatation (5%), and non-specific findings (11%). In support of the autoimmune theory, Barnes et al. showed that cholangiocytes, when cultured with RRV, expressed co-stimulatory molecules like B7-1 and B7-2 and produced pro-inflammatory cytokines and chemokines with a striking similarity to T-helper cells. Based on the premise that susceptibility to BA is ascribed to the lack of maternally derived immune protection, Turowski et al. tested oral vaccination with RotaTeq and Rotarix pre-conceptionally [96]. They found it could prevent RRV-induced BA in newborn mice. Despite this data, rotavirus is not a common etiology of BA in children. On the other hand, cholangiocytes remain susceptible to viral infection. Coots et al. immortalized H69 and primary human cholangiocytes infected with RRV, which showed a remarkable susceptibility [97]. This event supports a potential role of rotavirus infection in the pathogenesis of human BA like the murine models. Recently, Mohanty et al. described a rotavirus reassortant-induced rodent (mouse) model of hepatic fibrosis, which seems to impressively parallel human BA [98]. The authors of this investigation inoculated newborn pups with RRV, and this model of BA may help to illuminate some of the mechanistic aspects of the fibro-obliterative disease. The authors demonstrated that their T^R(VP2,VP4)^ model of murine fibrosis with bile duct proliferation can be used to study the mechanistic basis of BA. This model seems to be different from the CCl4 and bile duct ligation animal models. The histology portrayed in the publication shows findings that are relevant to human disease. Interestingly, transcriptome analysis substantiated the molecular pattern of genetic expression in this animal model with parallels to human BA. Reassortant viruses may be the key for future investigations. These models are created by coinfection of a cell with two parent strains that can produce progeny composed of various combinations of parental genes [98].


**Cell Culture-based Models**


Shiojri et al. [99] demonstrated the necessity of lung mesenchyme from four-day-old embryonic chicks. It was important to differentiate hepatocytes of the 9.5-day mouse embryo. Petersen et al. [100] reproduced bile ductile formation in vitro by culturing periportal mesenchyme with peripheral liver fragments from 15-day-old rat embryos (Carnegie Stage 21). Renal mesenchyme with liver fragments as a mesenchymal alternative showed a similar effect, but lung mesenchyme showed a weak inductive impact. These findings point to a possible alteration of mesenchymal cell migration or interaction that could lead to BA. This hypothesis has been confirmed in the human form of cholestasis secondary to DPM. On multispectral imaging, significant co-localization of the different markers in BA was detected in liver tissue, suggesting that the epithelial to mesenchymal transition occurs in liver fibrosis. In the future, and with major advancements, this animal model can be potentially used to investigate therapeutic interventions for BA.


**Toxin/Drug-induced Models**


The administration of drugs during pregnancy has also been shown to produce remarkable models of experimental BA in some animals both in pre- and postnatal ways.


**Wistar Rats**


The Wistar rat is a brown rat of the subspecies *Rattus norvegicus domesticus*, and photodynamic therapy (PDT) uses non-toxic dyes called photosensitizers (PSs), which absorb visible light to give the excited singlet state. In the presence of oxygen (O_2_), reactive oxygen species (ROS), such as singlet oxygen and hydroxyl radicals, are formed. Intraperitoneal phalloidin and PDT for Wistar rats have been used to create experimental BA. Hosoda et al. [101] administered intraperitoneal phalloidin (an actin-binding toxin) to pregnant Wistar rats. Histopathologic examination of the livers showed advanced fibrosis and thickening of the wall of the extrahepatic bile ducts and stenosis and near atresia of the ductal lumen. The liver showed interlobular fibrosis, but the complete obliteration, as in human BA, was found only in a few rats exposed to the drug during the fetal period.


**New Zealand White Rabbit**


In pregnant New Zealand (NZ) white rabbits, intravenous injection of lithocholic acid, a bile acid with cholestatic and inflammatory properties, produced obstruction within newborns’ biliary tract, which can be then diagnosed using ^99m^Tc-labelled compounds, which are excreted into the bile. However, there are no recent reports of such a drug-induced animal model for BA [102]. A variety of animal models of early neonatal cholestasis was reported in mice rats, mini-pigs, and hamsters by BDL or intra-biliary injection of sclerosants. These models seem to replicate the metabolic cholestasis, but other features of BA are lacking (Figure 1).


**Hamster**


Schmeling et al. [103] described an experimental obliterative cholangitis model of noninfectious biliary tract inflammation. An implantable osmotic pump was connected to a catheter placed into the gallbladder of golden hamsters (*Mesocricetus auratus*). The authors used phorbol myristate acetate (PMA). This compound is a diester of phorbol and a potent tumor promoter. It activates the signal transduction enzyme protein kinase C (PKC). PMA on PKC result from its similarity to one of the natural activators of classic PKC isoforms, diacylglycerol. In ROS biology, superoxide was identified as the major ROS induced by PMA but not by ionomycin in mouse macrophages. PMA has been routinely used as an inducer for endogenous superoxide production via stimulation of the mitogen-activated protein kinase (MAPK)/extracellular signal-regulated kinase (ERK) pathway [103]. PMA, when infused into the gallbladder, activates peribiliary neutrophils. Subsequently, there is inflammation and fibrosis without any potentially identifiable symptoms of obliteration.


**Biliatresone Animal Models**


During periods of drought there have been several outbreaks of BA in lambs and calves in Australia. The culprit was the ingestion of unusual plants by pregnant cows and ewes. The suspected plant seems originating from *Dysphania* plant species and biliatresone is the isoflavonoid caused damage to the biliary system. Both mice and zebrafish animal models have been realized. In zebrafish, the morphological disruption was found to be dose- and time-dependent. In the high-dose group, and between 8 and 13 days, the extrahepatic biliary system was difficult to distinguish, and the gallbladder had an abnormal morphology. The intrahepatic bile ducts were unaffected. A genetic mutant, labelled *ductbend*, showing an absence of intrahepatic bile ducts only, was treated with biliatresone. A synergistic effect was noted between the mutation and the toxin. Whole-exome sequencing revealed conserved areas of homology between the *ductbend* locus and a region close to *ductbend* to two well-known BA susceptibility loci [104]. A mouse model was recently established by a Shanghai group at the Children’s Hospital of Fudan University [105]. Neonatal rodents injected with biliatresone developed signs of biliary obstruction and dysplasia or the absence of extrahepatic bile duct lumina. Hepatic RNA-sequencing identified transcriptional evidence of oxidative stress.


**Genetic Models**



**Inv mouse model**


The *inversin* (*inv*) mutant mouse was created by insertional mutagenesis, and Japanese researchers [106] described *situs inversus* and severe jaundice in the *inv* homozygous mice mutants. A spontaneous mutation in the laterality *inv* gene on chromosome 4 between the *Tsha* and *Hxb* loci is present in these rodents. Homologs of these genes have been identified in humans as 6q21.1-q23; 118850 and 9q33; 187380, respectively. Obstructive jaundice and death usually occur within the first week of life. The intrahepatic bile duct and biliary system in the *inv* mice show markedly proliferative biliary ductules surrounding the portal vein like the DPM of some human BA [7]. DPM has also been observed in 20–40% of infants with BA with some prognostic relevance [7,29,31,32,34,107,108,109]. However, the extrahepatic bile duct does not show any evidence of ischemia or inflammation. These results indicate a congenital abnormal development of the intrahepatic biliary system in the *inv* mouse. The morphological changes are like the intrahepatic ducts with BA. Thus, the intrahepatic biliary system of the *inv* mouse could be an experimental BA model of BA with DPM. Alterations in the human orthologue of *Inversin* may contribute to those cases of BA in which *situs* anomalies are also observed. Despite this data, the absence of mutation and biliary anomalies in patients with lateralization defects demonstrates that *INV* may not be the only contributor to the etio-pathogenesis of BA in humans.

In the last few years, there have been some reports with other mouse models of genetic involvement, such as a mouse model with Sox17 haploinsufficiency published in 2013 and 2020 by Uemura et al. [54,110] and an animal model with Pkhd1 mutation published in 2018 by Huang et al. [111]. The SOX17-positive gallbladder/cystic-duct progenitor cells proliferate and thus are involved in the epithelial architecture of the gallbladder and cystic duct system. During the polarized proliferation and elongation processes of the extrahepatic biliary system along the proximodistal axis, some *Sox17*+/− epithelial cells present some defects in their epithelial maturation, exhibiting a detachment from the epithelial walls into the lumen, which is similar to an injury of the bile duct epithelial wall. The consequence is a congenital biliary atresia in extrahepatic ducts of *Sox17*+/− C57BL/6 embryos [54,110]. Regarding Pkhd1, the authors studied the nonobese diabetic (NOD) mouse model [111]. They previously reported that NOD.c3c4 mice harbor spontaneous autoimmune biliary disease with anti-mitochondrial antibodies, autoimmune histopathology, and T lymphocytes similar to human primary biliary cholangitis. In 2018, Huang et al. presented a novel NOD congenic mouse expressing aberrant polycystic kidney and hepatic disease 1 (*Pkhd1*) but lacking the c3 and c4 chromosomal regions (NOD.Abd3). This animal model showed lymphocytic infiltration and biliary duct epithelial hyperplasia (proliferation) with upregulation of genes involved in cholangiocyte injury/morphology and downregulation of immunoregulatory genes. This model is paramount for understanding loss of tolerance to biliary cells and may be highly relevant to the pathogenesis of several human cholangiopathies, including BA [111].

**Table 1 ijms-23-07838-t001:** Animal Models of Biliary Atresia.

Viral	Surgical	Toxin, Prenatal ^1^	Toxin, Postnatal ^2^	Genetic
**gpCMV** (Guinea pig): Infection-based => MNI, PC, Ch, F, BDP (Wang et al., 2011 [94])	**Post natal BDL** (Rabbit, lamb, rat, pig, monkey): Surgery-based => MNI, PC, Ch, F, ~ BDP (temp.) (Cameron & Oakley 1932 [79]) (Holder & Ashcraft 1966 [80]) (Morgan et al., 1966) (Spitz 1980 [81])	***Phalloidin*** (Wistar rat): **IP** Administration => Canalicular Ch, ↑ Vol. peri-canalicular actin filaments (Hosoda et al., 1997 [101]) ***1,4-phenylene-diisothio-*** ***cyanate*** (Wistar rat) **Oral** Administration => EHBDD, F, BDP (Ogawa et al., 1983 [112]) ***Monohydroxy bile acids*** (NZ white rabbit): **IV** Administration => BDO (in some offspring) (Jenner 1978 [102]) ***Biliatresone*** (BALB/c mouse): **IP** Administration => EHBD-A, MNI, F, BDP (Yang et al., 2020 [105])	***Phorbol myristate*** ***acetate*** (Golden hamsters): **GB** infusion => Peribiliary PMN, F (Schmeling et al., 1991 [103])	***inv mouse*** (OVE210 heterozygous ***inv*****mutant** mouse) IHBD-A (periportal), BDP, CBD patent (Shimadera et al., 2007 [106])
**RRV/HCR-3/WI-78** (BALB/c mouse): Infection -based => MNI, PC, Ch, ± F, -Atresia (Riepenhoff-Talty et al., 1993 [95])	**Obliterative micro-Sx** (Wistar rat): Microsurgery -based => F, BDP (zones 1 & 2) (Aller et al., 2004 [113,114])			**Sox17 haploinsufficiency** based mouse (C57BL/6 mouse) => Injury of the epithelial cells of the EHBDS, GB hypoplasia, BD stenosis/atresia (Uemura et al., 2013 [110]; Uemura et al., 2020 [54]))
**Reo Virus 3** (mice): Infection-based => MNI, PC, Ch (Phillips et al., 1969 [115])	**Transplantation** (C57BL/6 and B1O.A mice): Graft-based (Fetal/perinatal renal subcapsular allografts in adult congenic mice) => Fibrosclerosis (Schreiber et al., 1992 [89])			**Pkhd1-Nonobese** **diabetic (NOD)** mouse (NOD.*Abd3*) => MNI, BDP (Huang et al., 2018 [111])
**Rotavirus Reassortant–Induced Model** (RRRV: T^R(VP2,VP4)^) (Mouse): Infection-based => F, BDP (Mohanty et al., 2020 [98])	**Organ Culture** (Embryonal liver culture) Cell culture-based (BD Induction in embryonic liver) (Petersen et al., 2001 [100])			

Notes: ^1^ Prenatal (Wistar rat, NZ white rabbit, BALB/c mouse), ^2^ Postnatal (hamster, rat, mice, mini-pig). BD, bile ducts; CBD, common bile duct; IP, intraperitoneal; IV, intravenous; GB, gallbladder; IHBD-A, intrahepatic bile duct atresia (no patent lumen); EHBDD, extra-hepatic bile duct dilatation; F, fibrosis; MNI, mononuclear inflammation; NZ, New Zealand; PC, pericholangitis; PMN, polymorphonucleate leukocytes; Ch, cholestasis; BDP, bile duct proliferation/hyperplasia; RRV is a double-stranded RNA virus of the Reoviridae family composed of 11 gene segments encoding six structural (VP1-VP4, VP6, and VP7) and six non-structural (NSP1-NSP6) proteins. Pkhd1, polycystic kidney and hepatic disease 1.

## 3. Discussion

The most widely used experimental models are the RRV-induced murine model, BDL of mice, and inv mouse, but the biliatresone animal model is gaining in popularity (Figure 1). In the inv mouse model, the pathologic changes in DPM were found in the intrahepatic biliary system, which has been observed in some BA patients. The inv model has its limitations concerning the lack of inflammatory and immune-related symptoms and the fact that there is no parallel mutagenesis in laterality or *INV* gene homolog. Furthermore, BDL in rats requires a surgical procedure that needs extra time and expertise. Thus, currently, the biliatresone and the neonatal BALB/c mouse with intraperitoneal inoculation of RRV are very popular. The RRV mice model is consistent, easily constructible, and quick to realize, in that characteristic symptoms are evident as early as three days. The genetic inactivation of the interferon (*IFN*) gene in vivo completely prevented inflammatory obstruction of extrahepatic bile ducts in the resolution of jaundice and improved long-term outcome of mice challenged with RRV. However, there are inconsistencies by different groups in the presence of this virus in BA patients, and no study so far has managed to definitively prove the role of any specific hepatotropic virus as the etiologic agent of BA.

Talking of how far the animal models have narrated the untold story of mechanisms implicated in the pathogenesis of BA, we can point out specific salient facts. First, the RRV inoculated mice model of BA supports the feasibility of reproducing some features of BA. Second, the prevention of inflammatory bile duct obstruction in IFN-γ–deficient mice is evident of the immune dysfunction mechanism of BA. Despite the predominant T-helper cell response with increased IFN-γ–induced chemokines in the liver of humans and mice suffering from BA, there are numerous studies that fail to support viruses that BA triggers. Third, the inv mice model with bile duct obstruction and situs inversus with DPM and the findings of mutations in laterality genes (*CFC1*, *ZIC3*) in some patients with BA and laterality defects together underlines the role of morphogenetic defects in bile duct malformation. Fourth, increased biliary expression of caspases 1 and 4 and of IFN-γ–related and tumor necrosis factor-alpha (TNF-α)–related genes in neonatal mice with experimental BA on the one hand, and parallel studies in the liver from affected children showing increased expression of pro-apoptosis molecules on the other hand, underscore apoptosis as a significant mechanism of injury to duct epithelium in BA.

The expression of the polyductin in DPM of the intrahepatic biliary tree and in liver biopsies of patients with neonatal cholestasis, mostly BA, as well as in congenital hepatic fibrosis [4] and its re-expression during BA with DPM, underlines that polyductin expression should be studied in BA animal models. Polyductin can also be co-localized with liver progenitor cell markers like oval stem cell markers with some ductal plate cells during the normal and abnormal intrahepatic biliary system. Bile duct to portal space ratio (BD/PT) and ductal plate remnants can act as a valuable independent factor for the prognosis of neonatal liver disease of infants less than one year old [7].

Classical methodologies used to investigate the components of BA in experimental animal models include several approaches. They include dissecting and histopathological studies of liver or hepatobiliary tissues by immunohistochemical staining. In addition, Western blots, real-time PCR, adoption techniques (e.g., of the T cells from RRV-diseased mice into naïve syngeneic severe combined immunodeficiency recipients), apoptosis, and autophagy have been added. Pathological characteristics usually scored for BA in models involve liver tissue necrosis (inflammation, apoptosis scoring) and morphological changes in the biliary tract, such as bile duct inflammation, and anatomical or inflammatory profiles of portal/periportal regions. In addition, levels of total serum and conjugated bilirubin, fibrosis, liver functions, viral load, or replication in liver tissues are also considered. You et al. developed a genetic method to investigate the pathogenesis of BA. The gene expression profile of BA was downloaded from the Gene Expression Omnibus (GEO) database. It included 18 samples from newborn mice. These samples were collected at three time points after the induction of BA with rhesus rotavirus. The differentially expressed genes (DEGs) were identified using the limma package in R language, followed by hierarchical clustering analysis. The authors used gene ontology functional analysis and Kyoto Enrichment of Genes and Genomes pathway analysis of the selected common DEGs. Six common DEGs (*CCL3*, *CXCL5*, *CXCL13*, *CXCR2*, *CCL5*, and *CCL6*) were recognized that were involved in the significantly enriched functions and the significantly enriched pathways. These genes may be used for future experimental models of BA [100].

Overall, none of the models listed here may be acceptable as a unique and valid model for human BA. Experimental data from the various models in the future may need to be designed in a way to find hidden information common to multiple models, which is probably the “needle in the haystack” that might lead to a better understanding of the pathogenesis of human BA. Figure 1 highlights some of the most paramount models with the histopathology findings that the authors of this review could find in the original papers. The limitation of this study is that we did not have the original histological slides for pathological evaluation, but we relied on the text and the illustrative material published. Using standardized animal models may eliminate some differences between animal and human studies, allowing a greater degree of research translation. Liver transplantation teams face an impressive increase of individuals and families approaching liver transplantation worldwide. If the psychological and emotional burden to patients and families is considered, the quest for a cure is increasing exponentially. Although multiple etiologic factors have been repeatedly published in the literature, BA remains an orphan disease. In the 3rd decade of the 21st century, it seems that genomic profiling of BA is individualizing some factors that may be useful for understanding this disease. Nevertheless, BA remains quite heterogeneous and complex as a disease. The acknowledgment of the multifactorial causality will condition our pathways to identify the needle in the haystack.

## 4. Materials and Methods

In this narrative review, we present some animal models of experimental BA and ponder their applications and limitations considering the significant findings and progress made in the last 50 years. We propose a figure with all models assessing the histology for a straightforward comparison of the histopathology findings in each of these models. Such suggestions may be helpful to delineate new lines of research, potentially creating new eclectic models combining elements of several models. We searched several databases: Medline, PubMed, Herb-Med, and Cochrane Libraries (“biliary” AND “atresia” AND “animal model”). The language was restricted to English, and articles were selected for relevance.

## Figures and Tables

**Figure 1 ijms-23-07838-f001:**
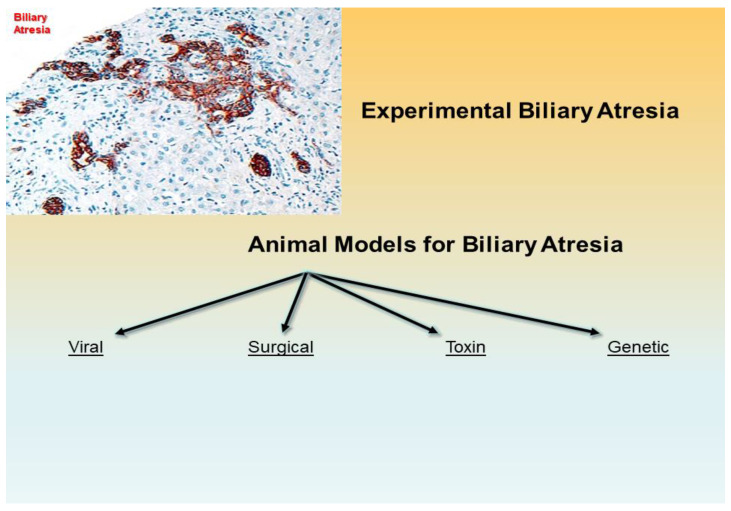
Experimental biliary atresia with fibrosis and bile duct proliferation of biliary epithelial cells in the portal tracts is evidenced by immunohistochemistry with an antibody against keratin 7 (intermediate filament of the cytoskeleton) (x200 as original magnification). An experimental biliary atresia may be setup with several laboratory animals with lines of scientific progress of this and last centuries toward viral, surgical, toxin-based, and genetic animal models.

## Data Availability

The data that support the findings of this study are available from the corresponding author upon reasonable request.

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
