# Peer review of "Biliary Atresia Animal Models: Is the Needle in a Haystack?"

_ijms, 2022, doi:10.3390/ijms23147838_

Round 1
Reviewer 1 Report
The authors provide a good summary of the modeling studies that have been published so far. The application of these models is necessary for disease research, but each model has certain drawbacks. How to combine the advantages of each model is more like piecing together pieces of a puzzle, which requires considerable exploration practice.
The following comments need to be further improved:
1. The contexts of line 62-94 and line 30-61 are repeated.
2. line 156. In Zebrafish section
HNF6, ARF6, Sox17, ILF2, DNMT1 genes have been confirmed to have effects on the development of bile ducts in zebrafish, and it is better to add in this section .
3. line 319. In Rotavirus section
The new RRV reassortant-induced murine model should be added in this section .
Author Response
Thank you for your comments and suggestions. We fully agree with you and added the missing information. We apologize for the delay, but we hope to have satisfied your criticisms.
The authors provide a good summary of the modeling studies that have been published so far. The application of these models is necessary for disease research, but each model has certain drawbacks. How to combine the advantages of each model is more like piecing together pieces of a puzzle, which requires considerable exploration practice.
The following comments need to be further improved:
1. The contexts of line 62-94 and
line 30-61
are repeated.
Thank you. The superfluous part has been deleted.
2. line 156.
In Zebrafish section HNF6, ARF6, Sox17, ILF2, DNMT1 genes have been confirmed to have effects on the development of bile ducts in zebrafish, and it is better to add them in this section.
Thank you. We added this data.
3. line 319.
In Rotavirus section
The new RRV reassortant-induced murine model should be added in this section.
Thank you. Done

Reviewer 2 Report
As stated in the manuscript, it is important to study for biliary atresia using appropriate animal models in order to understand the pathogenesis of the intractable disease and to improve therapeutic outcomes. The content of this article may provide useful information to readers.
However, there were some concerns that needed to be corrected for publication.
Overall, the text is redundant. For example, from page 2, line 19 to page 3, line 10 of the Introduction section is exactly the same as the description that precedes it.
I wonder whether the Results section of Culture models on page 9 is necessary for what you want to describe in this paper. Only in this part, you described in vitro experimental models. Furthermore, there are other in vitro studies of biliary atresia that have been reported.
Another concern is the part of genetic models. The authors described inv mutant mouse. This model is one of the animal models with genetic involvement. Recently, there were some articles with another mouse models with genetic involvement, such as a mouse model with Sox17 haploinsufficiency published in Development in 2013 by Uemura et al. and a model with Pkhd1 mutation published in The Journal of Immunology in 2018 by Huang et al.
Regarding Pkhd1, the authors described the need to elucidate this gene in animal models. I think it would be desirable to add these models to the results section.
In response to the above point, if corrections are made, Figure 1 needs to be corrected as well. In addition, I feel that Figure 1 is difficult to understand. I think it would be better to make it into a table for better visibility. Please consider this.
Author Response
Thank you for your suggestions and comments. We deleted the redundant portion of the manuscript, added more models as suggested by you and the other reviewer, and expanded the pkhd1 portion as suggested. Thank you for suggesting implementing an informative table. We agree and we added this table. Figure 1 is just now informative of the histopathology and the lines of progress for experimental biliary atresia.

Round 2
Reviewer 2 Report
The manuscript is well refined and I do not have further comments.